# Geographic Scene Understanding of High-Spatial-Resolution Remote Sensing Images: Methodological Trends and Current Challenges

**Peng Ye** [1,2,3,4], **Guowei Liu** [2,*] **and Yi Huang** [5,6]

1 Urban Planning and Development Institute, Yangzhou University, Yangzhou 225127, China; 007839@yzu.edu.cn
2 College of Architectural Science and Engineering, Yangzhou University, Yangzhou 225127, China
3 Key Lab of Virtual Geographic Environment, Ministry of Education, Nanjing Normal University, Nanjing 210023, China
4 Jiangsu Center for Collaborative Innovation in Geographical Information Resource Development and Application, Nanjing 210023, China
5 Smart Health Big Data Analysis and Location Services Engineering Lab of Jiangsu Province, Nanjing University of Posts and Telecommunications, Nanjing 210023, China; huangyi@njupt.edu.cn
6 School of Geographic and Biologic Information, Nanjing University of Posts and Telecommunications, Nanjing 210023, China
* Correspondence: arguoweiliu@yzu.edu.cn; Tel.: +86-155-2142-2026

**Abstract:** As one of the primary means of Earth observation, high-spatial-resolution remote sensing images can describe the geometry, texture and structure of objects in detail. It has become a research hotspot to recognize the semantic information of objects, analyze the semantic relationship between objects and then understand the more abstract geographic scenes in high-spatial-resolution remote sensing images. Based on the basic connotation of geographic scene understanding of high-spatial-resolution remote sensing images, this paper firstly summarizes the keystones in geographic scene understanding, such as various semantic hierarchies, complex spatial structures and limited labeled samples. Then, the achievements in the processing strategies and techniques of geographic scene understanding in recent years are reviewed from three layers: visual semantics, object semantics and concept semantics. On this basis, the new challenges in the research of geographic scene understanding of high-spatial-resolution remote sensing images are analyzed, and future research prospects have been proposed.

**Keywords:** geographic scene; high-spatial-resolution remote sensing image; scene understanding; semantic hierarchy of geographic scene; remote sensing image processing

## 1. Introduction

Remote sensing, as a comprehensive modern surveying and mapping technology, plays an important role in Earth observation. In recent years, as a result of the rapid development of sensor technology, aerospace platform technology and data communication technology, as well as the vigorous promotion of relevant international organizations, the global observation capability of the space–air–ground integration has been greatly enhanced [1]. At present, a large number of high-spatial-resolution (HSR) remote sensing images with meters, or even sub-meters, can be obtained. In HSR remote sensing images, various realistic geographic scenes are clearly presented: for instance, artificial construction scenes such as urban residential areas, ports and airports; disaster scenes such as landslides, mudslides and earthquakes; and natural scenes such as forests and beaches [2]. This small-scale observation means that HSR remote sensing images can provide more complex surface structure information and more sophisticated texture and size information. Consequently,

it has been applied to urban planning, disaster management, environmental monitoring, military activities and many other fields [3,4].

As a collection of multiple objects and their surroundings in the real world, understanding the semantics of scenes is an important task in remote sensing image interpretation. Scene understanding is based on the perception of remote sensing image data, combined with visual analysis, image processing, pattern recognition and other technical means, to mine the characteristics and patterns in the image from different levels such as computational statistics, behavioral cognition and semantics, so as to realize the effective analysis, cognition and representation of the scene. However, due to the limitations of space imaging technology, HSR remote sensing images, although of higher spatial resolution, are relatively deficient in spectral information [5]. In HSR remote sensing images, the spectral heterogeneity of the same type of ground objects is enhanced, and the spectral diversity of different ground objects is reduced, which leads to the decline in statistical separability of different types of ground objects in the spectral domain [6]. Therefore, understanding the geographic scenes in the HSR remote sensing images includes the identification of both objects and the relationships between objects, as well as the analysis of themes categories with richer concepts and content implied in the geographic scene. Because of the complexity and intersection of these tasks, the research on geographic scene understanding of HSR remote sensing images still faces many challenges, mainly including the following three aspects:

(1) In terms of the basic principles of geographic scene understanding, the machine will identify the objects or targets contained in the scene according to the similarity of image data. In contrast, humans analyze the semantic information of scene content through the category and spatial distribution of ground objects, and form high-level features through abstract concepts [7]. There is a semantic gap between the conceptual similarity of human understanding and the digital storage form similarity of machine identify. This makes it impossible to relate low-level visual features (such as color, shape, texture, etc.) to high-level semantic information directly.

(2) In terms of the data characteristics of HSR remote sensing images, the improved spatial resolution makes the ground objects in the images have more fine texture features, more obvious geometric structure and clearer location layout. Correspondingly, it also aggravates the difficulty of data processing in intelligent image interpretation. In high resolution images, the spectral heterogeneity of similar objects is enhanced, and the spectral difference of different objects is reduced. This leads to a decrease in the statistical separability of different ground objects in the spectral domain [8]. A high resolution does not necessarily promote an improvement in interpretation accuracy.

(3) In terms of the sophistication of geographic scenes, the structure and composition of the geographic scenes in the HSR remote sensing images are complex, highly variable and even messy. The types of geographic scenes with the same ground objects may be different. However, different types of ground objects also appear in similar geographic scenes [9]. Consequently, understanding the semantic information of the geographic scene and constructing the corresponding semantic feature description is crucial.

As an extension of remote sensing image interpretation, the complexity and comprehensiveness of geographic scene understanding based on HSR images is beyond the general processing task of remote sensing. Although significant progress has been made in the research of feature extraction, target detection, scene classification and other sub-tasks, these sub-tasks lack a unified framework to cross the "semantic gap" to understand the high-level semantics of the geographic scenes. Thus, it is necessary to integrate these sub-tasks according to the human cognitive model in understanding the geographic scenes of HSR remote sensing images. In recent years, many researchers who are engaged in computer vision have realized the importance of a "holistic understanding" of geographic scenes and put forward the research approaches of task integration and feature integration. However, there is no systematic research on the geographic scene of HSR remote sensing images as a comprehensive and complete field of intelligent information processing. This paper is focused on answering the following research questions:

(1) What are the objectives of geographic scene understanding?
(2) How are remote sensing approaches being used for geographic scene understanding?
(3) What are the current gaps in HSR remote-sensing-based geographic scene understanding?

The rest of the paper is organized as follows: Section 2 describes the basic ideas of geographic scene understanding based on HSR remote sensing images; Section 3 presents the semantic understanding approaches of the visual layer; Section 4 presents the semantic understanding approaches of the object layer; Section 5 presents the semantic understanding approaches of the concept layer; and Section 6 discusses the open problems and challenges in the future. The paper closes with a conclusion in Section 7.

## 2. Basic Ideas

The concept of geographic scene understanding of remote sensing images includes two aspects, namely, "remote sensing image understanding" and the "geographic scene". Remote sensing image understanding is a cognitive process to realize the objective things and their laws reflected by remote sensing images through observing, distinguishing, identifying and reasoning remote sensing images and interpreting the content of remote sensing images semantically. The geographic scene is a regional complex with a specific structure and function, which is composed of various natural and human factors in a certain region [10]. In remote sensing images, the geographic scene is a closed region composed of different ground objects. The geographic scene generally involves three aspects: (1) the constituent elements of the scene structure, (2) the relationship between these elements and (3) the function of the set of these elements. Therefore, the research object of geographic scene understanding of remote sensing images is the regional complex composed of ground objects with certain spatial distribution patterns. The research objective is to interpret the research object as a series of meaningful and understandable semantic information.

HSR remote sensing images can reflect the more detailed composition and spatial distribution of the ground objects, which is a microcosm of the real geographic scene. Owing to increased spatial resolution and unique imaging methods, geographic scenes in HSR remote sensing images have the following characteristics:

(1) The categories of ground objects in the geographic scene are diverse. The same category of geographic scene can contain different ground objects, and different geographic scenes can also contain the same ground objects. Different objects also have different characteristics in terms of spectrum, texture and structure [11].
(2) The categories of ground objects in the geographic scene have variability. The change in the categories of some ground objects in geographic scenes does not necessarily lead to a change in the whole semantic information of geographic scenes [12].
(3) The spatial relationship between ground objects in geographic scenes is complex. Different distribution forms between ground objects lead to different semantic information of geographic scenes. Other relevant characteristics are shown in Figure 1.

The characteristics of HSR remote sensing images also make the following special features in understanding geographic scenes:

(1) The semantic information of geographic scenes in HSR remote sensing images is hierarchical. The content description of HSR remote sensing images has the hierarchical inclusion relation of "Pixel-Region-Target-Scene". Different levels of image content reflect the semantic information with different levels of abstraction, which can be divided into the visual layer, object layer and concept layer (Figure 2). The visual layer is the description of pixel-level image content, including color, texture, shape and other original visual characteristics. The semantic information of the visual layer can be obtained directly from image processing without any external knowledge and experience [13]. The object layer is the description of region-level and target-level image content, including the individual features of objects and the local features of spatial relations among objects [14]. The semantic information of the object layer needs to be obtained through simple reasoning, and it is necessary to use external knowledge and experience to assist this reasoning. The concept layer is a description of the scene-level image content, including the abstract attributes of

the image. The semantic information of scene level involves the semantic features of scene representation or higher-level behavior or emotion analysis, and it needs to link image content with abstract concepts through complex reasoning and subjective judgment [15].

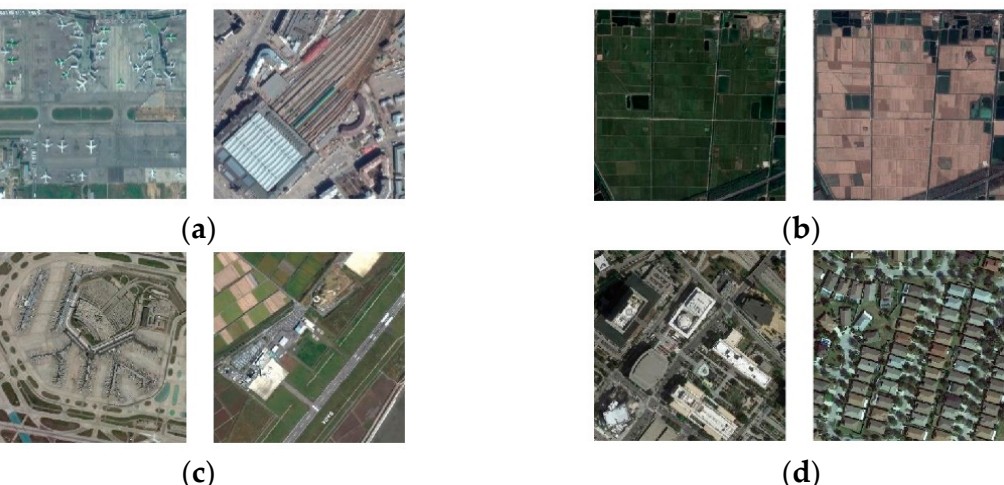

**Figure 1.** Image characteristics of geographic scenes in HSR remote sensing: (**a**) reflects the diversity of the ground objects in the geographic scene; (**b**) reflects the diversity of imaging conditions of HSR remote sensing images; (**c**) reflects the differences in the types of ground objects in the same category of geographic scenes; (**d**) reflects the similarity in the types of ground objects in different categories of geographic scenes.

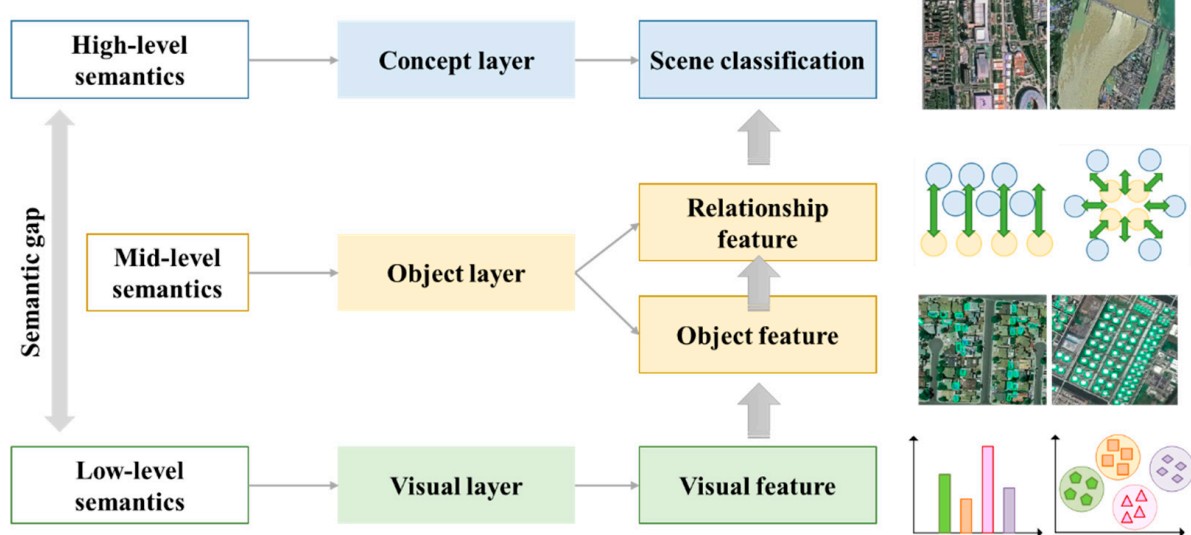

**Figure 2.** Semantic hierarchy model of three layers structure.

There is a dialectical relationship between the tasks of geographic scenes understanding at different semantic layers. The input of geographic scene understanding is the original HSR remote sensing images, and the output is semantic information of the geographic scene. In the tasks of geographic scene understanding, it is necessary to combine the semantic processing of the visual layer and object layer (feature extraction and target recognition) with the semantic reasoning of the concept layer (scene description and classification), the different tasks are interdependent on each other [16]. The cognition of visual and object layer semantics can form the inference of concept layer semantics, and the cognition of concept layer semantics can be used as knowledge to guide the extraction of visual and object layer semantics.

(2) Spatial structure characteristics play an important role in the geographic scene understanding of HSR remote sensing images. Because of the global and polysemy of the geographic scene, the geographic scene understanding is not a simple stacking of some local semantics [17]. In the same category of the geographic scene, the objects of the same type have similar individual characteristics and spatial distribution patterns. However, there are different structural features among different categories of geographic scenes. In Figure 3, these two geographic scenes of "residential area" and "industrial area" contain similar visual features and object types, which are composed of buildings, roads and vegetation. However, there are great differences in spatial structures between objects, which is a critical factor to distinguish the categories of geographic scenes. Therefore, the spatial structure characteristics of geographic scenes are relatively stable, and making full use of spatial information such as geometry, texture and context of HSR remote sensing images is an effective way to improve the understanding of geographic scenes [18,19].

**Figure 3.** Understanding differences of geographic scenes at different semantic layers.

(3) The data characteristics of HSR remote sensing images have both opportunities and challenges for the geographic scene understanding. The amount of HSR remote sensing images increases significantly. As the spatial resolution increases, the area of the ground covered by each pixel decreases significantly. This makes the ground object details and spatial distribution of HSR remote sensing images clearer. Compared with medium–low-resolution remote sensing images, HSR remote sensing images can be interpreted at the scene level, where semantic information is more abstract. However, compared with natural images, the HSR remote sensing images used for geographic scene understanding are less accessible in terms of data availability, except for the differences in shooting distance, shooting angle and imaging sensors. Natural images can be easily and quickly obtained from the Internet, and a large amount of data has been given the relevant label information when uploaded to the Internet [20]. For example, the ImageNet dataset [21,22] contains more than 14 million labeled samples in 1000 categories. HSR remote sensing images are not freely available for political, military and security reasons. HSR remote sensing images often rely on professional interpretation or even field research to obtain the correct label, the available sample size is limited. For instance, the UC-Merced dataset contains only 21 categories, with a total of 2100 labeled samples. Currently, the HSR remote sensing image datasets commonly used for geographic scene understanding are shown in Table 1.

**Table 1.** Comparison of existing public datasets.

| Datasets | Spatial Resolution (m) | Image Size | Number of Categories | Number of Samples per Category | Total Number of Samples | Year of Publication |
|---|---|---|---|---|---|---|
| UC-Merced [23] | 0.3 | 256 × 256 | 21 | 100 | 2100 | 2010 |
| WHU-RS19 [24] | 0.5 | 600 × 600 | 12 | 50 | 950 | 2010 |
| RSSCN7 [25] | - | 400 × 400 | 7 | 400 | 2800 | 2015 |
| RSC11 [26] | 0.2 | 512 × 512 | 11 | About 100 | 1232 | 2016 |
| SIRI-WHU [27] | 2 | 200 × 200 | 12 | 200 | 2400 | 2016 |
| NWPU-RESISC45 [28] | 0.2–30 | 256 × 256 | 45 | 700 | 31,500 | 2017 |
| PatternNet [29] | 0.062–4.693 | 256 × 256 | 38 | 800 | - | 2017 |
| AID [30] | - | - | 30 | - | 10,000 | 2017 |
| EuroSAT [31] | - | 64 × 64 | 10 | 2000–3000 | 27,000 | 2019 |

## 3. Semantic Understanding of Visual Layer

The semantic understanding of the visual layer of the geographic scene is the extraction of basic characteristics from remote sensing image data. The essence of geographic scene understanding is to establish the mapping relationship between low-level visual features and high-level scene semantics. Thus, extracting the visual features of HSR remote sensing images is the basis of the content description of geographic scenes, which includes local features and global features.

The global feature is the feature that can represent the whole image, and it has good invariance, simple calculation and intuitive representation. Common global features include color features, texture features and shape features. Among them, color features such as color sets, color moments, color correlation diagrams, color histograms [32] and color aggregation vectors [33–35] are insensitive to size and orientation and have good stability [36]. Texture features such as gray level co-occurrence matrix (GLCM) [37], the grayscale difference [38], autocorrelation function [39], gray-level run-length [40], local binary pattern (LBP) [41], etc., are characterized by local irregularity, but macroscopic regularity. Visual features cannot only describe the basic attributes of the image such as color, texture and shape, but also reflect the deep structure information of the image. In 2001, GIST was proposed by Aude Oliva et al. simulating human vision to roughly extract the image and its context information [42]. GIST can extract spectral information from the image globally as its representation without segmenting the image or detecting the target in advance. GIST is simple and easy to use. However, with the increasing complexity of image content and structure, such as the analysis granularity being too coarse to ignore the details of the objects in the scene, the result of image processing is far from the correct result. In general, the global features are sensitive to the actual imaging conditions, and the robustness and generalization ability are relatively poor.

The local feature can effectively resist various affine transformations and have some invariance. David Lowe has come up with a landmark local feature descriptor, the scale invariant feature transform (SIFT), which has good scale invariance and rotational invariance [43]. Thus, SIFT is one of the most widely used features in image processing. Bay et al. present an accelerated robust feature descriptor (SURF) inspired by SIFT [44]. While SURF is inferior to SIFT in scale scaling and rotational invariance, it is superior in blur and illumination variation and is several times faster than SIFT. With the advancement of research, the instability of color, light and gradient features in the process of recognition became an obstacle to image classification. The histograms of oriented gradients (HOG) feature proposed in 2005 continue the high recognition accuracy characteristic of the local feature; the gradient histogram method is used to effectively solve the problem of the low recognition rate of local scene contours due to the sensitivity of light and gradient features [45]. However, HOG features have high dimensions, low computational efficiency and great redundancy and do not consider the effect of scale transformation on classification results. The CENTRIST feature proposed by Wu et al. in 2010 solves this problem well [46]. Through the census transformation of the acquired pixels, these pixels are transformed into statistical histograms to form the CENTRIST feature to extract the object's local shape

structure. After the CENTRIST transformation, the image still retains the global and local structure information. Therefore, it can simulate the human visual system and describe the shape and texture of objects accurately.

Global features and local features have their own advantages and disadvantages. Different visual features are suitable for different tasks of geographic scene understanding. In HSR remote sensing image description, visual feature extraction should not only keep the invariance of features but should also fuse the spatial structure information of the features.

## 4. Semantic Understanding of Object Layer

Object layer semantics mainly describe the logical concepts of scenes in images, usually based on a large number of visual layer descriptors. Compared with the visual layer, the object layer is closer to the human understanding of the geographic scene. For instance, in the process of geographic scene understanding, we rely more on the houses and roads in the images, rather than recognizing that there are small dense and regular highlighted areas, narrow and long gray-banded areas and so on in the image. Houses, roads, sky and grass, which conform to human cognition, constitute the object layer semantics of geographic scenes. In addition, object layer semantics can also be abstract local areas, such as visual words generated by feature detection algorithms. There is also a certain context structure between different objects, forming a corresponding spatial relationship. Thus, a geographic scene is a combination of a set of specific objects. According to the different semantic forms of objects, there are three types: target object semantics, local area semantics and spatial structure semantics.

### 4.1. Target Object Semantics

The object layer semantics of the geographic scene are usually concentrated on the basic level of human cognition, which can be represented by many target objects (Figure 4). For the semantic understanding of the target object, it is necessary to use the target detection algorithm to clarify the types of each object. In the fields of computer vision and pattern recognition, many target detection algorithms have been developed, for instance: the threshold-based detection method [47], the template-based detection method [48], target detection based on Hough transform or Hough forest [49], target detection based on classifiers [50], etc. For the target detection of HSR remote sensing images, the method of target detection in the computer vision field is usually used for reference, and the research is carried out around the object of special interest, in particular, the artificial structures closely related to human activities, such as buildings [51], ports [52], airport runways [53], roads [54], warships [55] and so on. For artificial structures with obvious shape features, it is generally possible to directly use their unique shape features for detection, for instance, extracting straight lines to detect linear targets in images [56]. For complex targets, the corresponding models can be constructed; for instance, the "Building" target model can be constructed by texture, shape and SIFT features, and the "Port" target model can be constructed by combining the information of coastline, wharf and embankment [57].

For the target detection of HSR remote sensing images, it is more challenging to detect objects with large image sizes and various details. Target detection of HSR remote sensing images is studied from different perspectives. A multi-layer SVM classifier is used to exclude non-target regions to improve the speed of target detection in high-resolution remote sensing images [58]. The large remote sensing image is divided into smaller blocks, the salient and synopsis features of each block are extracted, and the target detection is realized by classification [59]. Target detection is also accomplished by first segmenting HSR remote sensing images and then merging regions related to the target based on knowledge [60]. In addition, the successful application of visual selective attention to the target location in large-format remote sensing images, and the results show that the visual attention mechanism can quickly focus on the place where the object to be detected appears in the complex large image. These methods are all beneficial explorations in the target detection of HSR remote sensing images [61].

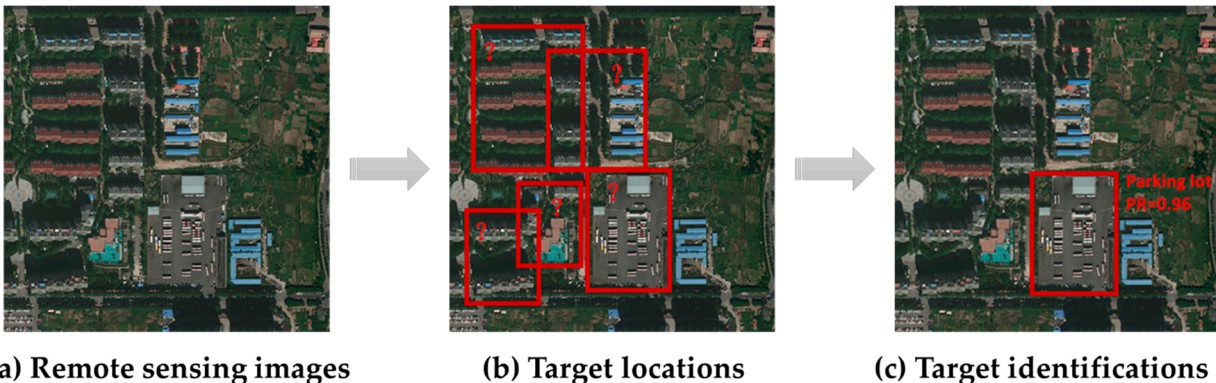

**Figure 4.** Target detection schematic. (**a**) is the original remote sensing image; (**b**) reflects that the candidate locations of the targets are found in the image; (**c**) reflects the results of target identifications.

The existing target detection methods have strong pertinence and lack universal and robust target detection models and algorithms. The motion characteristics of the target to be detected (such as ship wake, submarine track), the use of the shadow of the target in the image, the removal of the visible cloud cover and so on need to improve the target detection model with pertinence. To realize the practicality of target detection, it is necessary to establish a target detection model and a fast algorithm for multi-source data fusion.

### 4.2. Local Area Semantics

Remote sensing images can also be divided according to specific rules. By extracting the local image descriptor of each sub-block, the correspondence between the local descriptor and the local semantic concept is established, and the object layer semantics are extracted. Due to the differences of descriptors, feature extraction methods of the local area can be divided into three categories: visual dictionary, feature mapping and topic model.

### 4.2.1. Visual Dictionary

The visual dictionary, also known as the visual codebook, maps feature data onto individual codewords to generate feature vectors with codebook length [62]. The construction of a visual dictionary is essentially a cluster problem, and the visual codewords correspond to the cluster center. In the task of geographic scene understanding, the visual dictionary connects the image visual features with the scene semantics.

Whether the design of a visual dictionary is effective mainly includes three aspects: resolution, compactness and universality. (1) The resolution of the visual dictionary is reflected in the similarity between visual words. The lower the similarity, the higher the resolution. (2) The compactness is reflected in the choice of codebook length, which corresponds to different classification accuracies. A high recognition rate can be achieved by selecting a suitable visual dictionary. (3) The universality mainly refers to whether the visual dictionary needs to be relearned if the data of new categories are added. Existing dictionary learning includes generative (unsupervised) and discriminative (supervised) approaches. Perronnin et al. design a universal visual dictionary and a category visual dictionary to compete for the description of image content. The universal visual dictionary is used to describe all image scene classes, and the category visual dictionary for a certain scene class can be obtained by adaptive learning from the universal visual dictionary [63]. If an image belongs to a given class, a category visual dictionary is more suitable for describing the image than a general visual dictionary. On the contrary, the general visual dictionary is more suitable to describe the image than the category visual dictionary. However, traditional visual dictionaries are prone to a lack of clear meaning or polysemy. To solve the above problems, Su et al. use semantic attributes to clarify semantic meaning and integrate semantic attributes into the visual dictionary to remove the ambiguity of visual words [64].

### 4.2.2. Feature Mapping

After constructing the visual dictionary, it is necessary to encode and map the local features of the image, and to represent the semantic information of the image by transforming the local features into some organized form of visual words. In addition, there are some problems such as the low efficiency of dictionary generation, serious quantization errors and the lack of spatial information of visual words. Furthermore, the image semantic representation based on the visual dictionary is a linear representation, which only performs well in the case of the classifier with the nonlinear kernel, such as support vector machine (SVM). This will undoubtedly reduce its usefulness, making it difficult to apply to large-scale data set classifications. In recent years, a semantic representation based on feature mapping has attracted more attention. Feature mapping is used to quantize and code the visual features according to the visual words and generate the representation of the visual features in the visual dictionary.

Vector quantization (VQ) is simple and convenient, but its constraint conditions are too strict, resulting in the lack of information after visual feature quantization. To overcome this shortcoming, the sparse regularization approach can be used to loosen the constraints in the VQ, which translates into a sparse coding [65,66]. Sparse coding (SC) uses a sparse regularization method to reduce quantization errors and improve the uniqueness of feature coding. However, sparse coding is only a shallow learning model with a single hidden layer. The visual dictionary acquired by shallow learning lacks the selectivity of features, which will reduce the semantic resolution of image content. On the basis of SC, the local-constrained linear coding (LLC) is proposed [67]. The sparsity of feature coding cannot guarantee its locality, while the locality of feature coding can guarantee its sparsity. As a result, LLC is more efficient and has a better refactoring effect and local smooth sparsity [68].

### 4.2.3. Probabilistic Topic Model

In order to improve the performance of image semantic expression, a visual language model is proposed [69], which is inspired by the probabilistic topic model (PTM) of natural language understanding. Based on the visual language model, an image can be divided into many blocks as visual words according to certain rules, and these visual words have certain grammatical rules and spatial dependencies, also called visual grammar. The semantic information is represented by the co-occurrence frequency and spatial dependence of local features in the image. Common PTMs include probabilistic Latent Semantic Analysis (pLSA) [70] and Latent Dirichlet Allocation (LDA) [71].

To ameliorate the robustness of the visual language model to the change of target scale, Wu et al. extended the original model to multi-scale, and proposed the scale-invariant visual language model (m-VLM) [72]. Jing et al. use LDA to realize the scene classification of optical remote sensing images and compare it with the bag of visual words (BOVW) model [73]. The results show that LDA can provide more concise and abundant semantic information for image representation. In the parameter training stage, the probability of a visual language model is estimated by counting the frequency of a visual word or visual word combinations in the image. This approach equates the visual words in the target area of the image with the visual words in the background area, thus ignoring the negative impact of background noise on the target semantic representation [74]. Therefore, if we can distinguish the visual words in the background and assign the weight according to their contribution to the target, we can enhance the resolution of the visual language model to image semantic representation.

### 4.3. Spatial Structure Semantics

The different arrangements of the objects that comprise the geographic scene will make the geographic scene have a different spatial structure. Spatial structure information in HSR images is contained in spectral features and prior knowledge. For the understanding of spatial structure semantics, it is necessary to describe, model and extract them and obtain

a vector model representing the structural features of processing units (pixels, primitives and targets).

### 4.3.1. Pixel-Neighborhood-Window-Based Method

Taking a pixel as the basic processing unit, a window is defined for each pixel (also called the central pixel) in the image, which describes the spatial distribution pattern of the pixel values in the window area. The spatial structure features of the pixels in the window area are used as the spatial structure features of the central pixel [75]. This method describes the spatial structure of pixel neighborhoods. It can make up for the lack of spectral information of the central pixel by using the information of neighboring pixels, but it is important and uncertain for the reasonable selection of window size [76].

Among the existing methods, two kinds of neighborhood structure patterns are common. One is the interactive mode between the central pixel and its neighbor pixels, which is a "one-to-multiple" relationship. This relationship is represented using methods such as random fields, local spatial autocorrelation statistics and data fields [77,78]. The other is the spatial structure relationship of multiple pixels in the neighborhood window. The method equates the center pixel with its neighbor pixel, and is a "multiple-to-multiple" relationship, which is represented using methods such as the gray level co-occurrence matrix, global spatial autocorrelation statistic and the spatial semi-variogram function [79,80].

### 4.3.2. Object-Oriented Method

The basic unit of object-oriented processing is homogeneous objects (image blocks, homogeneous areas or patches) with certain semantic information in images [81]. The method needs to segment the image to obtain the objects to further describe the spatial structure of the objects in the images [82]. The advantage of this method is that it has more abundant spatial relationships for the objects themselves and is convenient for extracting spatial features [83,84]. The deficiency of this method lies in its serious dependence on the quality of image segmentation. In fact, inaccurate image segmentation results in error accumulation when understanding spatial structure semantics [85].

### 4.3.3. Rule-Partition-Based Method

The rule-partition-based method is similar to grid division. Firstly, the image is divided into regular (generally square) image blocks. Then, each image block is used as the processing unit to describe the spatial structure features of each image block [86]. This method is especially suitable for the detection of the spatial structure semantics of complex objects such as residential areas and aircraft. It does not focus on the detailed structure of objects in the image block but only on the statistical properties of the overall structure [87,88]. The deficiency of this method lies in how to determine the suitable partition of image blocks, especially when it cannot locate the object boundary accurately [89].

### 4.3.4. Global Organization Method Based on Local Structure

In this method, firstly, the local structural features, such as feature points, feature lines and feature surfaces, are obtained. Then, according to the spatial structure of the objects, the global structure model of the objects is constructed by using certain organization rules and mathematical models [90]. The process is mainly based on the geometric structure of the object itself, spatial relationship information and prior knowledge of the object structure [91]. For instance, when extracting building targets in HSR remote sensing images, we can make full use of the feature that the building roof is a rectangular structure. Firstly, local structure features such as corners, lines and ridges are extracted. Then, the method of perceptual organization is used to organize it into a complete roof contour of the building [92,93]. This method accords with the cognition rule of people to things, but it has a higher request for the construction of mathematical models and the realization of calculation methods [94].

## 5. Semantic Understanding of Concept Layer

The concept layer belongs to high-level abstract semantics. The concept layer semantics of the geographic scene is the comprehensive judgment and representation of concepts such as function and pattern. The main application of the remote sensing image processing method is scene classification. In general, high-level semantic information can be acquired based on low-level information analysis, and low-level information can be transferred to higher-level by modeling. Through layer-by-layer refinement, the final representation of the concept layer semantics is closer to the abstract thinking of human beings, and then the geographic scene semantics in the HSR remote sensing images have more practical significance. Therefore, the concept layer semantics of geographic scenes are derived from the visual layer semantics or the object layer semantics.

### 5.1. Visual Features Based Method

The concept layer semantics of the geographic scene can be directly described by low-level visual feature attributes. The scene classification algorithm based on visual features extracts the low-level visual features (such as color, shape and texture), then describes the features and designs the classifier to infer the semantic information of the geographic scene. According to the different sources of low-level feature extraction, scene classification based on low-level features includes two categories: global-feature-based methods and local-feature-based methods. The extraction methods of global features and local features in visual features are detailed in Section 3 (Table 2). Common classifiers used for visual features include maximum likelihood [95], minimum distance [96] and K-means clustering [97].

**Table 2.** Comparison of main methods in visual features.

| Name | Type | Output | Advantage | Disadvantage | Applicable |
|---|---|---|---|---|---|
| GIST | Global | Spectral information | Low computational complexity and easy to use | Poor performance in complex scenes with dense targets | Simple natural scenes |
| SIFT | Local | Neighborhood histogram | Suitable for translation, rotation, scale transformation | Poor performance in complex scenes with overall layout | Natural scenes |
| HOG | | Vector | Representation of contours and edges | Poor performance in scenes with unstable shape structure | Scenes with global structural stability |
| CENTRIST | | Census transformed value | Highlight local characteristics and reflect position information | Poor performance in complex and volatile scenes | Scenes with clear layout and sparse target distribution |

A single low-level visual feature is not suitable for the complicated task of geographic scene classification, and more methods of multi-feature fusion are applied. Feature fusion combines color, texture and other features into high-dimensional feature descriptors, and then uses a neural network to achieve feature dimension reduction [98]. In addition, on the basis of local features, the image is divided into local blocks, and the low-level visual features of each block are taken to establish the multi-feature fusion descriptors [99,100]. Nevertheless, the method based on local or global visual features and their fusion of visual features is not effective. The core problem is that the concept layer semantics need to infer from the low-level features to obtain the high-level semantic representation, while the visual-features-based method just lacks this semantic representation.

### 5.2. Object Semantics Based Method

In order to fully describe the complex characteristics of the geographic scene, the extraction method of concept layer semantics based on object semantics is widely used in geographic scene understanding of HSR remote sensing images. By extracting the local features in the geographic scene, the local features are mapped to the visual dictionary or parameter space to obtain more distinguishable object layer features. Then, these features

are input into the classifier to obtain the comprehensive description features of the whole geographic scene.

### 5.2.1. Target-Recognition-Based Method

Geographic scenes involve the interaction of many objects in complex semantic patterns. According to the experience of human visual perception, images containing similar objects may represent the same geographic scene. When defining the category of the geographic scene, different objects have different importances in the scene. This prior knowledge provides ideas for the classification of geographic scenes.

The method based on object recognition will identify the semantics of each object in the geographic scene and train the classifier for concept semantic understanding based on the semantic information of each object. Typical approaches include Object Bank [101], Latent Pyramidal Regions [102], Bag of Parts [103] and Latent Semantic Analysis [104] (Table 3). These approaches assume that a scene consists of a series of targets, and that by identifying and recognizing those targets with significant discrimination, the category of the scene can be inferred from the semantics of those targets [105]. In these approaches, the problem of semantic understanding of the concept layer is first transformed into the problem of target recognition, and then the geographic scene is represented by image blocks containing multiple targets. However, the errors caused by target recognition will further result in "error propagation", which will affect the semantic understanding of the geographic scene.

**Table 3.** Comparison of main methods in target recognition.

| Name | Advantage | Disadvantage | Applicable |
|---|---|---|---|
| Object Bank | Identifiable targets and natural scenes | High computational complexity and high feature dimension | Natural scenes with landmark targets |
| Latent Pyramidal Regions | Good performance for regions with specific structures | Focus on the shape structure of the scene, lack of deep semantic understanding | Scenes with complex background and crowded targets |
| Bag of Parts | Good performance for areas with boundaries or corners | | |
| Latent Semantic Analysis | The synonym is characterized by dimensionality reduction, and the redundant data are used | Polysemous words have low discrimination and high computational complexity | Scenes with heterogeneous information and clear boundaries |

### 5.2.2. Local Semantics Based Method

To avoid the process of object detection and recognition, the HSR remote sensing image can be divided according to rules and the local image descriptors of each sub-block can be extracted. The correspondence between local descriptors and local semantic concepts is established, and the scene classification is completed by using the probability distribution of local semantic concepts. There are two main algorithms based on local semantic concepts: the probabilistic topic-model-based method and the bag-of-visual-words-model-based method. Because the feature of spatial structure expresses the relationship between objects, it does not exist independently. Therefore, this feature is often used in conjunction with the bag of visual words model or the probabilistic topic model to enhance semantics.

(1)　Bag of visual words model

The visual codebook is defined in advance, and the image content is described by the probability distribution of the appearance of the visual codewords. Then, the geographic scenes are classified according to the probability distribution. In the process of constructing the bag of visual words model, feature extraction, visual dictionary learning, feature mapping and whether to add spatial context information all have an impact on the classification results [106].

In the aspect of feature extraction, we consider the construction of multi-feature scenes in low-dimensional space under different perspectives and use feature complementarity to carry out feature fusion to solve the problem of dimension reduction from a multi-

perspective [107]. Existing visual dictionary learning includes generation (unsupervised) and discriminant (supervised) methods [108,109]. After visual dictionary learning, the image local descriptor is mapped to the visual dictionary. The modification of the mapping method can improve the representation of local semantics [110]. In addition, feature description can be incorporated into feature space partitioning to capture the high-order structure inherent in the scene [111]. This makes the local semantics have both local gradient information, local structure information [112] and global spatial information [113–115].

(2)    Probabilistic topic model

The scene semantic content is first modeled by probability distribution based on the codewords. Then, the latent semantic topics in the images are learned by using the probability distribution model. In addition, the geographic scenes are classified according to the probability distribution of latent semantic topics.

Semantic topic modeling includes the generative probabilistic model and the discriminative probabilistic model. The generative probabilistic model of the geographic scene is constructed according to the joint probability distribution of the scene category in the feature space, using pLSA [116], LDA [117] and improved LDA (ts-LDA, css-LDA) to mine the latent semantic information of visual words [118]. Because various scenes contain different space-level structures, spatial information can undergo weighted fusion based on local semantic content. The discriminative probabilistic model is based on the conditional probability distribution of the category of the geographic scene in feature space, and its core task is to design kernel function. Wu demonstrates that a support vector machine based on a histogram intersection kernel (HIK) is more efficient than a radial basis function kernel for histogram-based data [119].

The generative probabilistic model and the discriminative probabilistic model have their respective advantages and complementary characteristics. The contradiction between computational complexity and model complexity is the biggest problem in the generative probabilistic model, but it is not a problem in the discriminative probabilistic model. The discriminative probabilistic model does not consider the connection between geographic scenes when modeling different categories of the scene, which belongs to independent modeling. In [120], these two probabilistic models are combined to complete the task of scene classification, and the classification effect is better than the single probabilistic model.

### 5.3. Feature-Learning-Based Method

Both the method based on visual features and object semantics rely mainly on artificial design in feature extraction, which is not only subjective, and it is not enough for more complex HSR remote sensing images. In recent years, feature learning, especially deep learning, has been introduced into the field of remote sensing for semantic understanding of geographic scenes due to its excellent performance in image classification [121]. The general flow of geographic scene classification based on feature learning is shown in Figure 5.

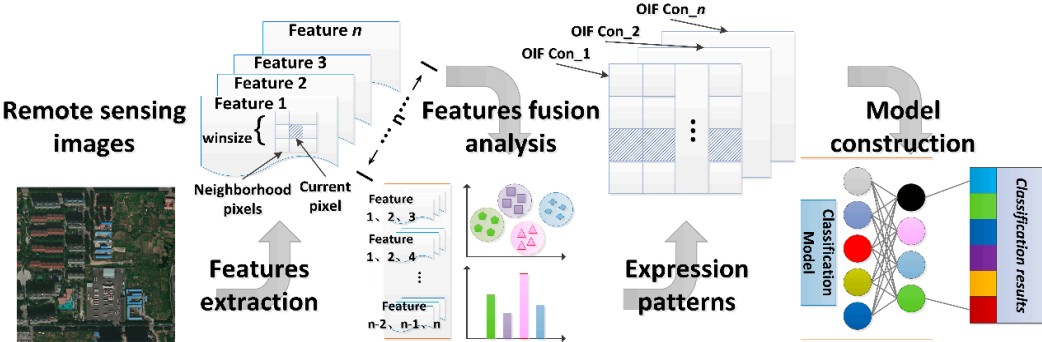

**Figure 5.** Geographic scene classification diagram.

Semantic understanding of the concept layer based on feature learning refers to the process of learning a potential scene classification feature through a series of mapping and transformation by using HSR remote sensing images as input of the model in machine learning tasks. Machine learning models can autonomously express and extract features from image data, abandoning the previous pattern of extracting features based on pre-designed rules [122,123]. Therefore, in the face of a complex surface environment, better classification results of geographic scenes can be obtained. At present, the commonly used machine learning models include sparse coding [124], neural network [125], support vector machine [126] and deep learning [127] (Figure 6). As a new intelligent method of pattern recognition in recent years, a deep learning network composed of multilaminate nonlinear mapping layers has become an especially important development direction in the field of remote sensing image processing [128]. Deep Learning is a deep structure neural network, which can extract the features of remote sensing images better than shallow structure models such as artificial neural networks and support vector machine models. Moreover, deep learning models can learn more abstract and distinguishable semantic features autonomously. The deep learning approach converts the semantic understanding of the concept layer into an end-to-end problem. On the one hand, the pre-trained deep learning network structure can be directly used to learn the global features in the visual layer of images to understand the semantics of the concept layer [129]. On the other hand, the deep learning network can also be used as a local feature extraction operator to jointly complete the semantic understanding of the concept layer with the help of feature code technology. Common deep learning models include convolutional neural networks (CNN) [130–132], deep belief network (DBN) [133], recurrent neural network (RNN) [134], automatic encoders [135], graph convolutional networks (GCN) [136], generative adversarial networks (GANs) [137,138] and so on. The deep learning method can be divided into three categories according to the supervision mode: (1) full supervision, (2) semi-supervised and (3) weak supervision.

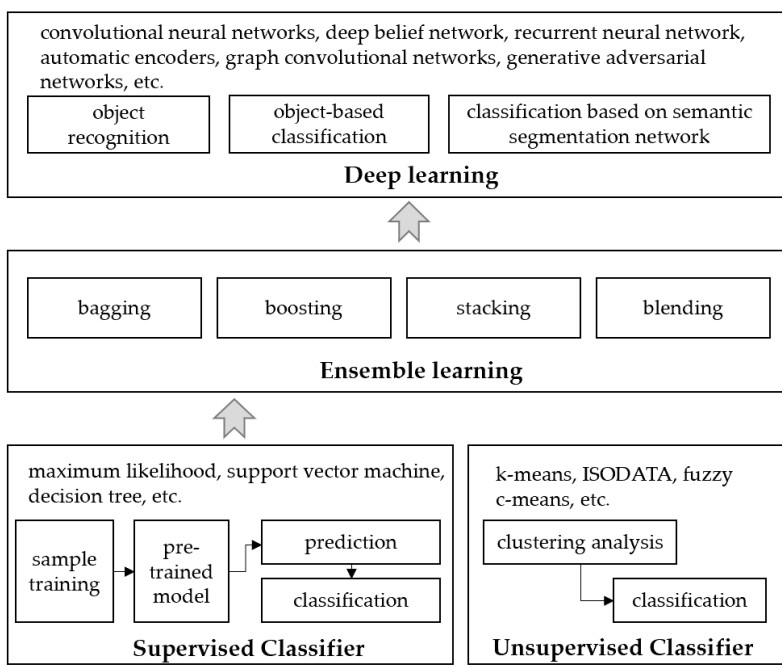

**Figure 6.** Development of feature learning methods.

### 5.3.1. The Method Based on Fully Supervised Deep Learning

Nowadays, most geographic scene classifications of HSR remote sensing images based on deep learning can be classified as full supervision. The integration of multiple learning models is one of the ways to improve the learning effect. Zhu et al. [139] proposed an adaptive deep sparse semantic modeling (ADSSM), which combines the topic model with

CNN and effectively integrates sparse topic features and deep features at the semantic level. Cheng et al. [140] proposed a new loss function to train fused deep neural networks by combining deep learning with metric learning. Zhang et al. [141] combined CNN and CapsNet for scene classification. This approach combines the advantages of both networks while leveraging the powerful feature extraction capabilities of CNN and the excellent feature fusion and classification capabilities of CapsNet. He et al. [142] proposed a new skip-connected covariance network (SCCov) for remote sensing image scene classification. Sumbel et al. [143] presented the BigEarthNet, which is a new large-scale, multi-label Sentinel-2 benchmark archive. The experimental results obtained in the framework of scene classification problems show that a shallow CNN architecture trained on the BigEarthNet provides much higher accuracy compared to a state-of-the-art CNN model pre-trained on the ImageNet. Thus, the BigEarthNet opens up promising directions to advance operational remote sensing applications and research in massive Sentinel-2 image archives.

### 5.3.2. The Method Based on Semi-Supervised Deep Learning

Semi-supervised learning can make use of a large number of unlabeled samples, reducing the need for labeled samples, which, to some extent, solves the problem of insufficient labeled samples in the field of deep learning [144]. Han et al. [145] proposed a generic framework based on semi-supervised deep features from the perspective of expanding the scale of labeled samples. In this framework, multiple support vector machine (SVM) models are applied to the label recognition of easily confused category samples, which improves the label precision and the number of labeled samples, thus improving the generalization ability and classification precision of the network.

It is also an effective semi-supervised deep learning to construct a feature extraction model based on unsupervised learning in the feature learning stage, then train the classifier with labeled samples. Soto et al. [146] used a combination of labeled and unlabeled samples to train generative adversarial networks (GAN) and then used the trained classifiers for scene classification. At this point, the classifier has a large number of unlabeled samples of information, which is helpful to improve the final classification effect. Fan et al. [147] used the representative salient regions extracted from the image as unlabeled samples to train the feature extractor. Then, the extractor is used to extract the features of the samples to be classified. Finally, SVM is used to classify the extracted features.

### 5.3.3. The Method Based on Weak Supervised Deep Learning

In HSR remote sensing image scene classification tasks, weak supervision usually uses labeled samples similar to target samples to train scene classification models. This method divides the dataset into the source domain and target domain. The former is different from the latter but similar. The latter can obtain labels through various transfer learning and further be used for training scene classification models. Othman et al. [148] took the features extracted from labeled images as the source domain, and the features extracted from unlabeled images as the target domain. Then, apply them to network training and optimize the specified loss function to classify labeled and unlabeled data. Gong et al. [149] further improved deep structural metric learning (DSML) by proposing Diversity-Promoting-DSML (D-DSML), which reduces the parameter redundancy produced by DSML and improves the feature representation ability.

Some existing deep learning classification tools include OverFeat [150], DeCAF [151], Caffe [152], AlexNet and so on (Figure 7). However, in these models, learning millions of network parameters also requires millions of training data as input. In order to reduce the over-fitting problem, a smaller network structure can be constructed. However, the generalization ability of the network model trained by this method is limited, such as gradient enhancement convolutional neural network [153] and multi-perspective convolutional neural network [154]. Therefore, the unsupervised feature-learning method directly uses the network model trained on the data set of images as the feature extractor to extract the deep features of the image directly, or after the feature transformation is input into the clas-

sifier for classification, higher classification results could be obtained [155]. Furthermore, the stacked covariance pooling method transforms the extracted multi-layer convolution layer features to obtain the global deep features of the image, which can effectively fuse the multi-layer deep features [156].

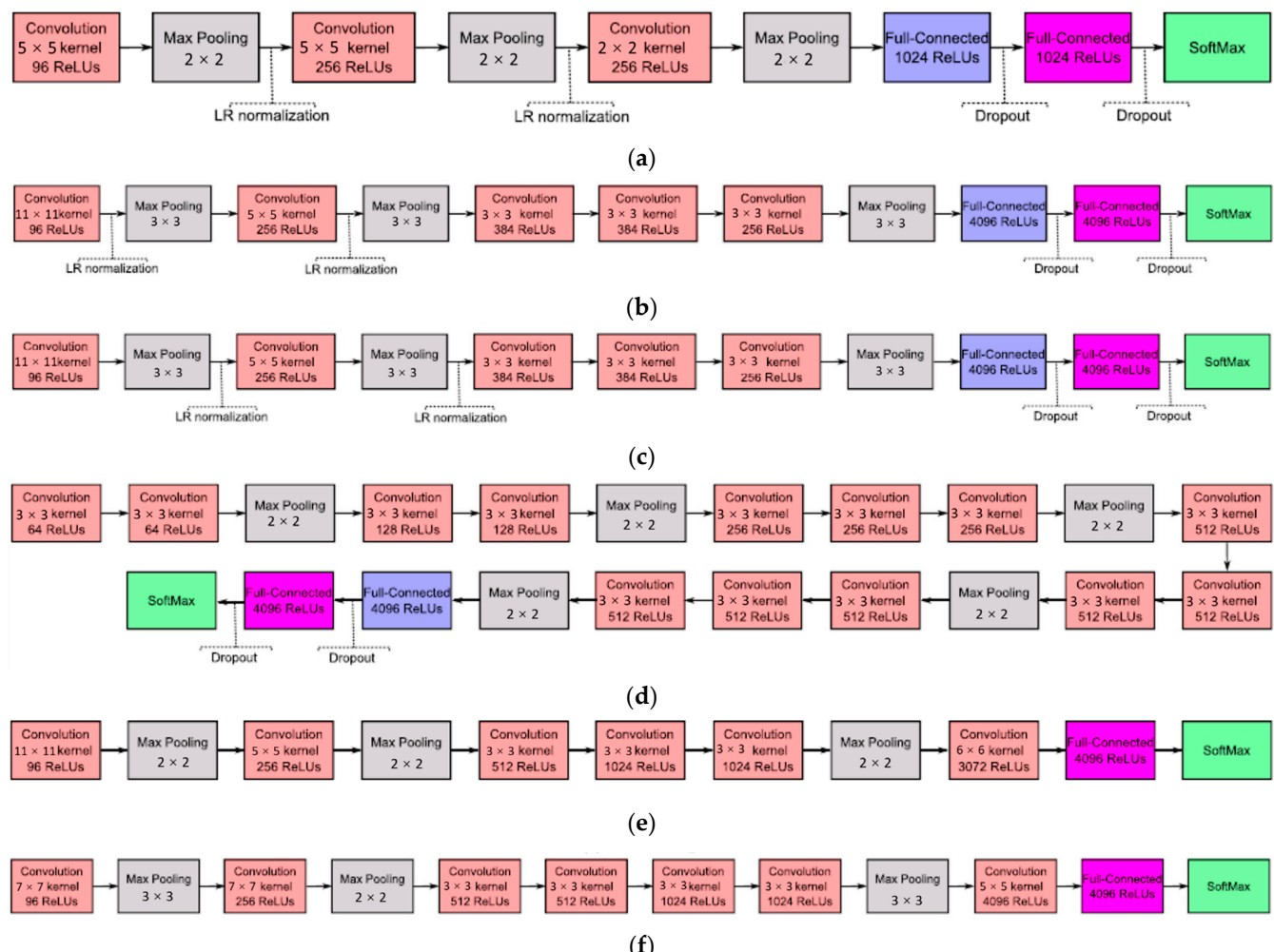

**Figure 7.** Architectures of different ConvNets evaluated in [157]. Purple boxes indicate the layers from where features were extracted in the case of using the ConvNets as feature extractors. (**a**) PatreoNet; (**b**) AlexNet; (**c**) CaffeNet; (**d**) VGG$_{16}$; (**e**) OverFeat$_S$; (**f**) OverFeat$_L$.

The method of deep feature fusion can further improve the accuracy of geographic scene classification. The most direct way is to cascade the features of the fully connected layers extracted from different network models [158]. In addition, discriminant correlation analysis (DCA) can be used to fuse the features of different fully connected layers [159]. Alternatively, the classification results of multiple models are fused based on Choquet fuzzy integral [160]. The UC-Merced dataset is one of the classic open resource sets for classification tasks of geographic scenes. Using the UC-Merced dataset as experimental data, the performance of existing geographic scene classification models is summarized (Table 4).

**Table 4.** Comparison of main methods in geographic scene classification.

| Method | | Accuracy (%) | Other Indicators |
|---|---|---|---|
| Visual features-based method | Gabor texture [161] | 76.91 | - |
| | Color-HLS [161] | 81.19 | - |
| | NN-STSIM [162] | 86 | - |
| | Quaternion orientation difference [163] | 85.48 ± 1.02 | - |
| | MS-CLBP [164] | 90.6 ± 1.4 | - |
| Object-semantics-based method | BoVW [161] | 76.81 | - |
| | BoVW + SCK [161] | 77.71 | - |
| | SPM [161] | 75.29 | - |
| | SPCK ++ [165] | 77.38 | - |
| | HMFF [166] | 92.38 ± 0.62 | - |
| | CCM-BoVW [167] | 86.64 ± 0.81 | - |
| | Wavelet BoVW [168] | 87.38 ± 1.27 | - |
| | UFL [169] | 81.67 ± 1.23 | - |
| | COPD [170] | 91.33 ± 1.11 | - |
| | FV [171] | 93.8 | - |
| | VLAT [171] | 94.3 | - |
| | SG-UFL [172] | 82.72 ± 1.18 | - |
| | PSR [173] | 89.1 | - |
| | UFL-SC [174] | 90.26 ± 1.51 | - |
| | SAL-PLSA [175] | 87.62 | - |
| | SAL-LDA [175] | 88.33 | - |
| Feature-learning-based method | CaffeNet finetune [176] | 95.48 | - |
| | GoogleNet finetune [176] | 97.1 | - |
| | Multiview DL [177] | 93.48 ± 0.82 | 84.35 (Sensitivity), 91.72 (Specificity) |
| | GBRCN [178] | 94.53 | - |
| | ADPM [179] | 94.86 | - |
| | HCSAE [180] | 97.14 ± 1.19 | - |
| | MARTA GANs [181] | 94.86 ± 0.80 | - |
| | Fusion by addition [182] | 97.42 ± 1.79 | - |
| | salM$^3$LBP-CLM [183] | 95.75 ± 0.80 | - |
| | TEX-Nets [184] | 97.72 | - |
| | CCP-Net [185] | 97.52 ± 0.97 | - |
| | CNN (LOFs+GCFs) [186] | 99.00 ± 0.35 | - |
| | ARCNet-VGG16 [187] | 99.12 ± 0.40 | - |
| | D-CNN with VGG16 [188] | 98.93 ± 0.10 | - |
| | SAL-TS-Net [189] | 98.90 ± 0.95 | - |
| | Two-stream deep fusion [190] | 98.02 ± 1.03 | - |
| | PMS [191] | 98.81 | $8.32 \times 10^6$ (Number of neurons) |
| | SSF-AlexNet [192] | 92.43 ± 0.46 | - |
| | VGG16+MSCP+MRA [193] | 98.40 ± 0.34 | - |
| | MCNN [194] | 96.66 ± 0.90 | - |
| | Bidirectional adaptive feature fusion [195] | 95.48 | - |

Although great progress has been made in geographic scene classification using deep learning algorithms, compared with the shallow algorithm, the classification effect has been improved obviously. However, the application of deep learning still faces many problems, such as the following:

(1) In terms of training data, the success of a deep neural network is that it can fit large-scale samples without sacrificing generalization ability. In the field of geographic

scene understanding, it is difficult to construct a large-scale, high-quality and complete HSR remote sensing image dataset for training. Firstly, from the perspective of time, a training sample can only represent the sampling of a time section. However, the interpretations of objects are dynamic in different periods. This time heterogeneity puts forward higher requirements for the quality, scale and completeness of sample annotation [196]. Secondly, from the perspective of space, due to the differences in climate and light conditions, the distribution of ground objects in different geographic scenes has natural heterogeneity [197]. This spatial heterogeneity leads to the imbalance of sample categories in the supervised learning process, whether within the training set or between the training set and the test set, which leads to "over-fitting" or "under-fitting" problems.

(2)  In terms of learning mechanisms, supervised learning mainly relies on semantic support provided by manual annotation as the only learning signal for model training. If human labeling is regarded as prior knowledge, the machine has been limited in knowledge in the process of labeling [198]. However, for the huge amount of image data, the intrinsic information should be much more abundant than the semantic information provided by sparse labels. Therefore, over-reliance on a manual annotation will cause the risk of "inductive bias" in the trained model. Moreover, the computational cost is high, especially for small samples. Most of the deep learning models are trained on the established network structure and are then fine-tuned to obtain better network parameters. This training pattern is not suitable for ever-expanding datasets.

Although deep learning has a strong learning ability, compared with real artificial intelligence, it still lacks the ability of abstract knowledge representation, reasoning causality and logical relationship. Therefore, there is a long distance to understand the geographic scene automatically through feature learning.

## 6. Open Problems and Challenges

(1)  Integrated system engineering for geographic scene understanding

The research of HSR remote sensing images is often only used the visual information and a little semantic information, such as target detection, image classification, image segmentation, scene classification and so on. These researchers can often only detect a certain target contained in the image, or obtain the category labels of each pixel or the whole image, but they do not make full use of the features of the image. Thus, it is difficult to mine the attributes, characteristics and relationships among the objects in the image in detail. In this way, images are not fully understood at the semantic level and HSR remote sensing data are not fully utilized. The organic integration of single subtask or feature information of HSR remote sensing image processing can enhance the performance of understanding, and it is more suitable for people's understanding mode of the geographic scene. The multi-class feature information and multi-subtasks are not completely independent, and the mutual influence and restriction factors should be considered comprehensively (Figure 8). Therefore, the construction of system engineering for geographic scene understanding can follow the following Formula (1):

$$Y = \{(feature_1 \oplus feature_2 \oplus \ldots \oplus feature_n) \otimes (task_1 \oplus task_2 \oplus \ldots \oplus task_n)\} \quad (1)$$

In the formula, $Y$ is the system engineering for geographic scene understanding, $\otimes$ and $\oplus$ represent the different combinations of features and tasks.

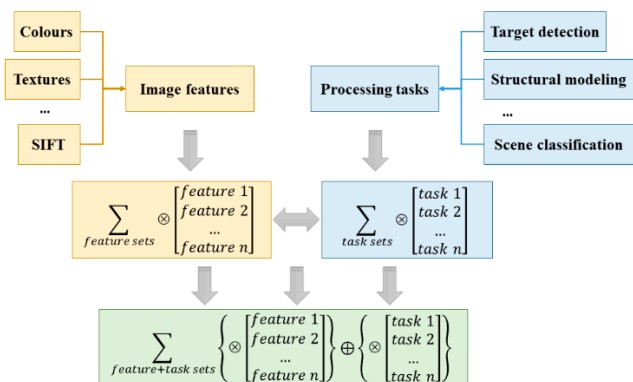

**Figure 8.** System engineering framework for geographic scene understanding based on HSR remote sensing image.

(2)  Comprehensive semantic representation of the geographic scene in HSR remote sensing image

The purpose of geographic scene understanding based on HSR remote sensing image is to semantically explain the content at all layers in the geographic scene. It is necessary to construct a comprehensive semantic representation model of geographic scenes, and to standardize and integrate the various layers and types of semantic information obtained from the understanding of geographic scenes. The semantic parsing tree of a geographic scene can be constructed, and the tree structure of and/or a graph can be used to represent the semantic content of understanding. The semantic parsing tree of the geographic scene follows a unified semantic specification, which is generally divided into four levels "scene-object-part-pixel" (Figure 9). The "And" node represents the decomposition, such as "scene→object", "object→part" and so on, which is followed the syntactic rules of "A→BCD". Any geographic scene semantics can be represented by this parsing tree structure, and the semantic hierarchy of geographic scene is clearly divided, which has both semantic attributes and semantic relations between different levels.

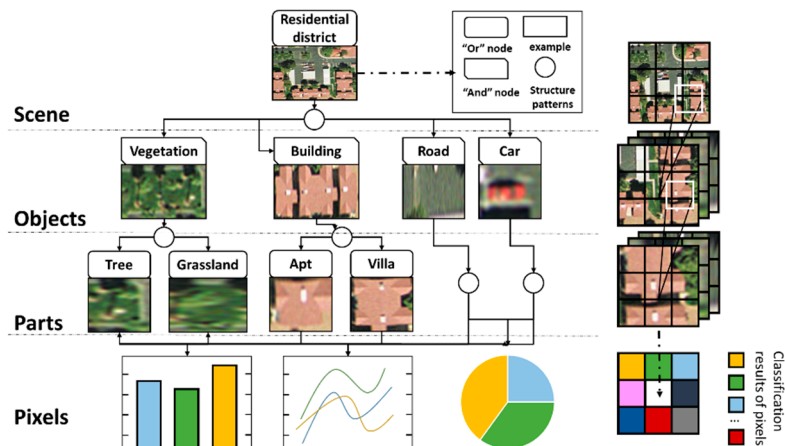

**Figure 9.** Hierarchical representation structure of the semantic of geographic scene.

(3)  Adaptability for large-scale complex geographic scenes

With the massive growth of image data and the continuous subdivision of scene categories, the problem of geographic scene understanding is faced with unprecedented challenges both in image quantity and scene category. The understanding of real geographic scenes requires higher complexity and depth than scene classification, and the solution to this problem will have a profound impact on artificial intelligence technology. For the current semantic understanding approaches of geographic scenes, there are still the

following problems to be solved: (1) The semantic extraction ability of the visual layer is insufficient. Even considering multiple visual features, most of them are the simple superposition of different features. (2) The semantic modeling of object layer is redundant and lacks homogeneity in the description, which makes it difficult to take into account the computational efficiency and effect. (3) The semantic understanding of concept layer ignores spatial location information and globality. Whether it is visual-feature-based or object-sematic-based method, it is difficult to obtain an accurate global description of spatial location relationships and geographic scene characteristics at the same time. Thus, it is not conducive to an accurate understanding of the geographic scene.

(4)　Fusion application of abundant multi-source data

In the big data era, the accessibility of various types of data has been broken through, creating conditions for the geographic scene understanding. On the one hand, the promotion of big data technology in the field of remote sensing has promoted the arrival of the era of remote sensing big data. Multi-source remote sensing data collaboration can integrate the advantages of various remote sensing observation methods and make up for the insufficient single sensor, which is one of the important research directions for the breakthrough of remote sensing image processing. On the other hand, the remote sensing image data record the natural environment of the surface, but the perception of changes in the social environment is scarce. The fusion of multi-source data is not only limited to HSR remote sensing data itself, but also needs to combine different types of data resources to make up for the deficiency of remote sensing monitoring. Social media data represented by Twitter, Facebook, Sina Weibo and Internet maps represented by GeoNames, GNIS, OpenStreetMap, etc., have become important sources of data for depicting the scenes of humanities and society. The fusion of HSR remote sensing images and multi-source data provides a new idea and method for geographic scene understanding.

## 7. Conclusions

Geographic scene understanding is one of the core tasks for middle and high-level cognition in remote sensing image processing tasks. Its complexity and comprehensiveness make it difficult to accurately understand the semantic information of geographic scenes. Based on the analysis of the basic concepts and core connotations of geographic scene understanding, this paper reviews the research status of geographic scene understanding from the tasks of different semantic layers in HSR remote sensing images. Geographic scene understanding decomposes the information of HSR remote sensing images into three semantic layers: based on the visual features of remote sensing images, the local objects, spatial structure and scene functions of the geographic scene are analyzed in a consistent cognitive system. This not only conforms to the logic and order of human cognition but also has significant interpretability of various semantic information. In terms of target detection, efficient and accurate feature representation and fusion of appropriate attention mechanisms are the core of extracting object category semantics. In terms of spatial structure description, pixel neighborhood window, object-oriented, rule partition and local structure are the main methods for extracting spatial structure semantics. In terms of scene classification, according to the semantic abstraction degree of extracted features, it mainly includes the visual feature classification method, object semantic classification method and feature learning classification method.

In future research, it is necessary to deeply study the intrinsic objective laws of various objects, textures, spaces and other information in geographic scene understanding, in order to reveal the relationship and influence mechanism between various features of images and different subtasks of image processing. The system engineering of geographic scene understanding is constructed from the global perspective, and the deep mechanism of human understanding of the geographic scene is explored. This is not only conducive to improving the adaptability of large-scale complex geographic scenes, but it also provides a universal cognitive structure for other HSR remote sensing images processing tasks such as image analysis and landscape investigation.

**Author Contributions:** Conceptualization, P.Y. and Y.H.; methodology, P.Y. and G.L.; validation, G.L.; formal analysis, P.Y.; investigation, P.Y. and G.L.; writing—original draft preparation, P.Y.; writing—review and editing, P.Y. and Y.H.; visualization, G.L.; supervision, G.L.; project administration, P.Y.; funding acquisition, P.Y. All authors have read and agreed to the published version of the manuscript.

**Funding:** This research was funded by the Open Foundation of Key Laboratory of Virtual Geographic Environment (Nanjing Normal University), the Ministry of Education (grant nos. 2021VGE01, and 2022VGE01), the Humanities and Social Sciences Foundation of Yangzhou University (grant no. xjj2021-08), the Open Foundation of Research Institute of Central Jiangsu Development, Yangzhou University (grant no. szfz202114) and the Open Foundation of Smart Health Big Data Analysis and Location Services Engineering Lab of Jiangsu Province (grant no. SHEL221 002).

**Institutional Review Board Statement:** Not applicable.

**Informed Consent Statement:** Not applicable.

**Data Availability Statement:** Not applicable.

**Acknowledgments:** The authors thank Xueying Zhang and Chunju Zhang for their critical reviews and constructive comments.

**Conflicts of Interest:** The authors declare no conflict of interest.

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
