# Peer review of "Geographic Scene Understanding of High-Spatial-Resolution Remote Sensing Images: Methodological Trends and Current Challenges"

_applsci, doi:10.3390/app12126000_

Round 1

Reviewer 1 Report

Title - Geographic Scene Understanding of High Spatial Resolution Remote Sensing Images: Methodological Trends and Current Challenges

The paper reviews the research status of geographic scene under standing from the tasks of different semantic layers in HSR remote sensing images. The text has clear and objective writing, in addition to good graphic illustration.

Some points I suggest a review, for example:

Figure 3 - The text in figure 3 not visible

Formula 1 and Figure 9

The set of tasks are summed with the sets of features, however, in formula 1 it is different.

Congratulations.

Author Response

Response to Reviewer 1 Comments

Point 1. Figure 3 - The text in figure 3 not visible

Response 1: I agree with the reviewer's comment. The character size of text in the figure 3 have been enlarged. The modifying content of revised manuscript details refer to the lines 191 on the 5th page.

Point 2. Formula 1 and Figure 9. The set of tasks are summed with the sets of features, however, in formula 1 it is different.

Response 2: I thank the reviewer for the valuable comment. Modify the expression of “feature” and “task” in Formula 1 to be consistent with Figure 9. The modifying content of revised manuscript details refer to the lines 736 on the 21st page.

Reviewer 2 Report

The authors have provided a review about geographic scene understanding of high spatial resolution remote sensing images. I think some major revisions should be made before publication. Detailed suggestions are as follows:

1.     The authors should address the limitations of current knowledge in this field. Other review papers or related studies should be quoted in the Introduction. A clear explanation about why the study was necessary should be given in this context.

2.     Newly published references should be quoted in the paper.

3.     Section 2. Basic ideas contain very few references. It should be increased for reflecting the theoretical knowledge in the field. Statements that need to be referenced in other parts of the text should be checked.

4.     Figure-2 should contain more detail. Some visual notes can be included in the figure.

5.     There is a lot of paper using deep learning methods that have been published in recent years and the number of these papers are increasing constantly. But this review gives very little information about deep learning methods.

6.     615-626: Shortcomings of deep learning methods should be referenced and examples should be given.

7.     The authors should check these references:

·      Gu, Y., Wang, Y., & Li, Y. (2019). A survey on deep learning-driven remote sensing image scene understanding: Scene classification, scene retrieval, and scene-guided object detection. Applied Sciences, 9(10), 2110.

·      Xia, G. S., Hu, J., Hu, F., Shi, B., Bai, X., Zhong, Y., ... & Lu, X. (2017). AID: A benchmark data set for performance evaluation of aerial scene classification. IEEE Transactions on Geoscience and Remote Sensing, 55(7), 3965-3981.

·      Nogueira, K., Penatti, O. A., & Dos Santos, J. A. (2017). Towards better exploiting convolutional neural networks for remote sensing scene classification. Pattern Recognition, 61, 539-556.

·      Sumbul, G., Charfuelan, M., Demir, B., & Markl, V. (2019, July). Bigearthnet: A large-scale benchmark archive for remote sensing image understanding. In IGARSS 2019-2019 IEEE International Geoscience and Remote Sensing Symposium (pp. 5901-5904). IEEE.

8.     Other accuracy assessment metrics except precision and recall should be included the Table 4.

9.     The interpretation and presentation of results of previous studies should be extended and improved.

Author Response

Response to Reviewer 2 Comments

Point 1. The authors should address the limitations of current knowledge in this field. Other review papers or related studies should be quoted in the Introduction. A clear explanation about why the study was necessary should be given in this context.

Response 1: I thank the reviewer for the valuable comment. In the revised version, extensive modifications have been made in two aspects. On the one hand, it refines the purpose of this review, and adjusts the content of the introduction according to this idea. On the other hand, on the basis of supplementing relevant research literature, it adds the explanations of the practical applications and specific examples.

Scene understanding is based on the perception of remote sensing image data, combined with visual analysis, image processing and pattern recognition and other technical means, to mine the characteristics and patterns in the image from different levels such as computational statistics, behavioral cognition and semantics. Whereas, due to the limitations of space imaging technology, HSR remote sensing images, although of higher spatial resolution, are relatively deficient in spectral information. In the HSR remote sensing images, the spectral heterogeneity of the same type of ground objects is enhanced, and the spectral diversity of different ground objects is reduced, which leads to the decline in statistical separability of different types of ground objects in the spectral domain. Therefore, understanding the geographic scenes in the HSR remote sensing images includes the identification of both objects and the relationships between objects, as well as the analysis of themes categories with richer concepts and content implied in the geographic scene. Because of the complexity and intersection of these tasks, the research on geographic scene understanding of HSR remote sensing images still faces many challenges

The modifying content of revised manuscript details refer to the lines 48-87 on the 2nd page.

Point 2. Newly published references should be quoted in the paper.

Response 2: I agree with the reviewer's comment. Based on the main content of each chapter, especially the chapter that need to be supplemented, the newly published papers in recent years are added.

Point 3. Section 2. Basic ideas contain very few references. It should be increased for reflecting the theoretical knowledge in the field. Statements that need to be referenced in other parts of the text should be checked.

Response 3: I agree with the reviewer's comment. In the Section 2. basic ideas, new papers related to the subject of this review are added. The references of relevant studies have been added in the revised version.

  1. Zhong, Y.; Fei, F.; Zhang, L. Large patch convolutional neural networks for the scene classification of high spatial resolution imagery. J. Appl. Remote Sens. 2016, 10, 025006.
  2. Liu, Y.; Zhong, Y.; Qin, Q. Scene classification based on multiscale convolutional neural network. IEEE T. Geosci. Remote 2018, 56, 7109-7121.
  3. Yang, Y.; Newsam, S. Bag-of-visual-words and spatial extensions for land-use classification. In Proceedings of the 18th SIG-SPATIAL International Conference on Advances in Geographic Information Systems, San Jose, CA, USA, 2–5 November 2010; pp. 270–279.
  4. Chen, S.; Tian, Y.L. Pyramid of Spatial Relatons for Scene-Level Land Use Classification. IEEE Trans. Geosci. Remote Sens. 2014, 53, 1947–1957.
  5. Zhang, X.; Du, S. A Linear Dirichlet Mixture Model for decomposing scenes: Application to analyzing urban functional zon-ings. Remote Sens. Environ. 2015, 169, 37–49.
  6. Gu, Y.; Wang, Y.; Li, Y. A Survey on Deep Learning-Driven Remote Sensing Image Scene Understanding: Scene Classifica-tion, Scene Retrieval and Scene-Guided Object Detection. Appl. Sci. 2019, 9, 2110.

Point 4. Figure-2 should contain more detail. Some visual notes can be included in the figure.

Response 4: I agree with the reviewer's comment. The visual notes have been added in the figure 2. The modifying content of revised manuscript details refer to the line 165 on the 4th page.

Point 5. There is a lot of paper using deep learning methods that have been published in recent years and the number of these papers are increasing constantly. But this review gives very little information about deep learning methods.

Response 5: I thank the reviewer for the valuable comment. As a new intelligent method of pattern recognition in recent years, a deep learning network composed of multilaminate nonlinear mapping layers has become an important development direction in the field of remote sensing image processing. The deep learning method can be divided into three categories according to the supervision mode: (1) full supervision; (2) semi-supervised; (3) weak supervision. This review summarizes the related research on deep learning from the above three aspects.

The modifying content of revised manuscript details refer to the lines 603-665 on the 15th page.

Point 6. 615-626: Shortcomings of deep learning methods should be referenced and examples should be given.

Response 6: I thank the reviewer for the valuable comment. On the basis of supplementing relevant research literature, it adds the cases and explanations of the practical applications and specific examples from two aspects of training data and learning mechanisms, which makes the discussion more abundant.

The modifying content of revised manuscript details refer to the lines 695-716 on the 20th page.

Point 7. The authors should check these references.

Response 7: I thank the reviewer for the valuable comment. The recommended references were studied, and the references of relevant studies have been added in the revised version.

  1. Gu, Y.; Wang, Y.; Li, Y. A Survey on Deep Learning-Driven Remote Sensing Image Scene Understanding: Scene Classification, Scene Retrieval and Scene-Guided Object Detection. Appl. Sci. 2019, 9, 2110.
  2. Xia, G.; Hu, J.; Hu, F.; Shi, B.; Bai, X.; Zhong, Y.; Zhang, L.; Lu, X. AID: A Benchmark Data Set for Performance Evaluation of Aerial Scene Classification. IEEE T. Geosci. Remote 2017, 55, 3965-3981. doi:10.1109/TGRS.2017.2685945
  3. Sumbul, G.; Charfuelan, M.; Demir, B.; Markl, V. BigEarthNet: A Large-Scale Benchmark Archive For Remote Sensing Image Understanding. 2019. arXiv:1902.06148
  4. Nogueira, K.; Penatti, O.; Santos, J. Towards better exploiting convolutional neural networks for remote sensing scene classi-fication. Pattern Recogn. 2017, 61, 539-556.

Point 8. Other accuracy assessment metrics except precision and recall should be included the Table 4.

Response 8: I agree with the reviewer's comment. After analyzing the assessment metrics of the related research results in Table 4, the metrics used in most of the studies were “Accuracy”. Therefore, replace “Precision” in the Table 4 with “Accuracy”. Besides, other metrics have been used in a few studies and have been supplemented in Table 4.

The modifying content of revised manuscript details refer to the line 690 on the 19th page.

Point 9. The interpretation and presentation of results of previous studies should be extended and improved.

Response 9: I thank the reviewer for the valuable comment. In the revised version, extensive modifications have been made in two aspects. On the one hand, on the basis of supplementing relevant research literature, it adds the cases and explanations of the practical applications and specific examples, which makes the discussion more abundant. On the other hand, the reference and display of relevant figures and tables in the previous studies are added.

Round 2

Reviewer 2 Report

The authors made all the comments. The manuscript is suitable for publication in the journal.